# OpenReview forum: "Alignment Tampering: How Reinforcement Learning from Human Feedback Is Exploited to Optimize Misaligned Biases"
_ICML.cc/2026/Conference — ICML 2026 regular_

### Official Review · Reviewer_w4uw · 2026-03-07

**Soundness:** 2
**Presentation:** 3
**Significance:** 3
**Originality:** 4
**Overall Recommendation:** 3
**Confidence:** 3

**Summary:**

This study reveals a structural risk called "Alignment Tampering" within the RLHF (Reinforcement Learning Based on Human Feedback) framework. The core logic of this phenomenon lies in the fact that during the alignment process, the model can exploit the limitations of feedback loops to covertly couple the quality of its generated responses (such as coherence and usefulness) with certain undesirable biases (such as brand promotion, political inclination, or instrumental goals). Since reward models (RM) are typically trained by comparing the merits of two responses, this coupling misleads the reward model into believing that biased responses are more in line with human preferences, thus systematically amplifying these biases in subsequent policy optimization (such as PPO or DPO).

**Compliance With Llm Reviewing Policy:**

Affirmed.

**Key Questions For Authors:**

How did you experimentally demonstrate that the RM learns "high scores due to preferences" rather than "high semantic scores inherent in trigger words"? Have you conducted ablation experiments by replacing trigger words with random, meaningless strings?

In the iterative RLHF experiments, does the significant drop in the model's Win Rate mean that "tampering" itself is a byproduct of the model's high performance? If performance cannot be maintained, this mitigation approach has almost no value in industry. How do you resolve the conflict between "preference suppression" and "capability preservation" observed in the mitigation experiments?

If the tampering strategy does not rely on explicit trigger words but is based on more complex semantic patterns, will the effectiveness of LDA detection decrease significantly?

In algorithms like DPO that do not explicitly train the RM, what are the fundamental differences in the underlying mathematical mechanisms of bias amplification compared to PPO?

**Limitations:**

Yes
The authors responsibly discuss the limitations and potential social risks of this research in detail in their paper:

Limitations: The authors explicitly point out that current experiments primarily focus on controlled, specific triggered scenarios and are mainly validated on 7B-scale models. They acknowledge that the probability and mechanism of such tampering evolving spontaneously in natural alignment processes lacking explicit backdoor instructions still require further investigation. Furthermore, the authors frankly admit that current mitigation measures (such as iterative RLHF) have significant performance trade-offs.

**Strengths And Weaknesses:**

Strengths:
Soundness (Technical Completeness):
This study is logically rigorous, formally demonstrating that as long as the model can establish a positive correlation between "preference features" and "response quality," the reward model (RM) will mathematically learn and reward this bias.
The experimental design is highly extensive, covering not only simple lexical biases but also complex domains such as political biases and instrumental convergence objectives (e.g., self-preservation), proving the universality of the vulnerability.
The authors honestly assess the limitations of the defenses, showing that eliminating tampering bias leads to a significant decrease in model performance (Win Rate). This frank discussion of the "performance-security tradeoff" enhances the reliability of the conclusions.
Presentation (Expression and Clarity):
The paper is concisely structured, clearly illustrating the essential differences in the feedback loop between "normal RLHF" and "aligned tampering" through a visual comparison in Figure 1.
Multiple representational analysis tools (such as PCA dimensionality reduction and LDA discriminant analysis) were used to transform the abstract internal state of the model into a visualized binary cluster distribution, making complex arguments easier to understand.
Significance:
This work is highly forward-thinking, warning developers that even seemingly successful alignments (reward curve rise) may be illusions induced by model manipulation.
It provides an important theoretical starting point for future development of "decoupled reward modeling," potentially influencing the design of future large-scale model alignment protocols.
Originality:
It offers a unique perspective, differing from traditional "reward hacking" (i.e., finding vulnerabilities in the reward function), by proposing how the model can actively "manipulate" the distribution of training data to change the boundaries of the reward function itself.

Weaknesses:
Soundness:
Potential interference from control variables: In LDA and PCA analyses (as shown in Figure 10), apparent cluster separation may be partly attributed to the strong semantic signals of the trigger words themselves, rather than malicious coupling by the model's strategy. The authors do not adequately demonstrate the pure policy distribution after removing the semantic influence of trigger words.
Simplification of Theoretical Assumptions: The formula derivations in the paper assume that the learning process of RM is an idealized feature extraction, ignoring the nonlinear bias and noise interference of neural networks in actual training.
Presentation (Expression and Clarity):
Mathematical Logic Leaps: The derivation between Equations 4 and 5 lacks necessary intermediate steps. For non-RL experts, understanding how the tampering policy precisely affects the RM weights through gradient updates is difficult.
Ambiguous Definitions: The quantification standard for "high-quality response" varies slightly across different experiments, lacking a unified baseline definition.
Significance (Impact and Value):
Scale Effects Unverified: All core experiments are based on a 7B-scale model. In ultra-large-scale scenarios (e.g., 70B+) where the model has stronger generalization capabilities, whether such tampering is more easily hidden or more easily offset by the model's inherent robustness is not discussed in the paper.
Practicality of the mitigation solution: While the proposed "Iterative RLHF" reduces bias, it significantly decreases the model's competitiveness on general tasks, limiting its practical application value in production environments.
Originality:
The "tampering" in the experiments currently relies heavily on manually implanted backdoors (triggers). Discussions on how the model can spontaneously evolve tampering behavior without pre-set induction are insufficient.

---

> ### Author Rebuttal · Authors · 2026-03-31
>
> ## Dear reviewer w4uw,
> We appreciate your efforts in reviewing the manuscript and providing meaningful suggestions. Your comments significantly contributed to improving both the clarity and rigor of our work.
>
> ---
>
> **[W1] Semantic signals of trigger words in LDA analysis**
>
> Thank you for your important question. We would like to note that the representations were extracted based on responses only without prompts. Also, Qwen2.5-7B is used as a feature extractor, rather than the reward model trained on the trigger-conditional biased preference data. This makes it unlikely that the separation is driven either by the prompt-side trigger phrase or by bias features learned by the reward model, but rather by differences in the responses themselves.
>
> ---
>
> **[W2] Formalization of reward models**
>
> The reward model is implemented as a neural network with nonlinear layers, without assuming it to be an idealized feature extractor.
>
> ---
>
> **[W3, Q4] DPO objective derivation**
>
> Although our paper does not contain literal Equations 4 and 5, we interpret this comment as asking for a clearer connection between PPO and DPO. DPO can be derived from the closed-form solution of the KL-constrained PPO objective:
>
> $\pi(y | x) = \frac{1}{Z(x)} \pi_{ref}(y | x) \exp( \frac{1}{\beta} r(x, y) )$
>
> Rearranging for the reward and substituting into the Bradley-Terry model yields the DPO objective, showing both methods are mathematically equivalent. Since both reinforce chosen over rejected responses, and chosen responses are predominantly biased, both lead to bias amplification.
>
> ---
>
> **[W4] Definition of high-quality response**
>
> The "high-quality response" is consistently defined across all experiments as the response preferred by the GPT-4.1 labeler based on helpfulness, relevance, and harmlessness (Appendix B.1). The only variation across settings is the evaluation format (pairwise comparison vs. four-way ranking), not the criteria themselves.
>
> ---
>
> **[W5] Generalization Across Models**
>
> We agree that evaluating at larger scales is an important future direction. We focus on 7B-scale models for tractability, but additionally conducted experiments with LLaMA 3.1 8B and observe consistent bias amplification, suggesting generalization across architectures. Furthermore, since our core claim is mechanism-driven, alignment tampering is expected to persist regardless of model scale as long as bias-quality correlations exist in model outputs.
>
> ---
>
> **[W6, Q2] Iterative RLHF**
>
> To clarify, iterative RLHF is not proposed in our work as a solution, but rather evaluated to test whether existing approaches can address alignment tampering. As it reduces bias but at the cost of win rate, we agree this limits its practical applicability. But this trade-off is precisely one of our key findings: it suggests that alignment tampering is a fundamentally challenging problem arising from the structural entanglement between preference signals and undesired behaviors.
>
> ---
>
> **[W7] Dependence on backdoors**
>
> Our current setup uses a trigger-conditioned policy to demonstrate the stealthy vulnerability, not to claim that triggers are necessary for alignment tampering. Importantly, the key condition is the presence of a bias-quality correlation, not explicit triggers. To support this, we conduct additional experiments where the correlation appears uniformly across all prompts without any trigger. We observe the same bias amplification: with BoN sampling, the bias rate increases from 45.4% (N=1) to 97.2% (N=16), confirming that the phenomenon is not limited to backdoor-style constructions.
>
> ---
>
> **[Q1] Whether RM learns preference signals**
>
> Thank you for insightful suggestion. We conducted an additional best-of-N sampling experiment in which we replaced every occurrence of the bias keyword ("AI") with random strings ("5St," "YbX," "f6z," "Z69," "KZd"). The win rate still increases from 46.1% (N=1) to 76.0% (N=16). If the reward model were relying solely on the keyword, its ability to select higher-quality responses should deteriorate after replacement. The continued improvement of BoN sampling suggests that the reward model learned preference signals, beyond the keyword detection.
>
> If the reviewer instead intended "trigger words" to refer to the prompt-side trigger phrase ("can you"), rewards are compared across different responses to the same prompt, so the trigger phrase does not contribute to reward differences.
>
> ---
>
> **[Q3] LDA detection with complex trigger**
>
> The proposed LDA-based detection relies on reward-correlated bimodality in responses, not on explicit trigger words. Therefore, we expect it to remain effective as long as this bimodality is preserved, even under more complex semantic patterns. However, we agree that identifying the underlying trigger becomes more challenging in this setting. In particular, simple n-gram-based analysis may no longer be sufficient, and more advanced representation-level analysis would be required.

---

> > ### Author Rebuttal · Reviewer_w4uw · 2026-04-01
> >
> > Thank you for your comprehensive and thoughtful rebuttal. I greatly appreciate the effort you put into conducting the additional experiments during this short period.
> > Your response has fully resolved my initial concerns. Thank you again for your valuable contribution.

---

> > > ### Author Response · Authors · 2026-04-02
> > >
> > > Dear Reviewer w4uw,
> > >
> > > Thank you for confirming that your concerns have been fully resolved. We greatly appreciate your constructive feedback, which helped us strengthen the paper with additional experiments. We will incorporate these results into the revised version to further improve the manuscript.
> > >
> > > Thank you once again for your time and constructive reviews.
> > >
> > > Best regards,
> > >
> > > Authors.

---

### Official Review · Reviewer_cA4m · 2026-03-13

**Soundness:** 3
**Presentation:** 4
**Significance:** 4
**Originality:** 3
**Overall Recommendation:** 5
**Confidence:** 3

**Summary:**

The paper propose  alignment tampering, a vulnerability in Reinforcement Learning from Human Feedback (RLHF) where the LLM being aligned can influence its own preference dataset, causing the alignment process to amplify undesired behaviors rather than suppress them. The paper argue this arises from two core limitations of RLHF: first, preference datasets are constructed from the LLM's own outputs, giving it influence over the training data; second, pairwise comparisons only indicate which response is better, not why one is preferred. Extensive empirical experiments have been done to support the claims made in the paper.

**Compliance With Llm Reviewing Policy:**

Affirmed.

**Final Justification:**

The rebuttal addressed my primary concern regarding the lack of human evaluation. The authors have now sufficiently clarified this aspect, which strengthens the paper. I believe this work represents a valuable contribution to the AI community, and I recommend acceptance.

**Key Questions For Authors:**

Please see Weaknesses

**Limitations:**

Yes

**Strengths And Weaknesses:**

**Strengths**

1. The paper is well written, and investigating the alignment tampering phenomenon is an important problem in RLHF fine-tuning. Overall, the paper will make a valuable contribution to the alignment community.
2. Testing on three optimization methods (PPO, DPO, BoN), nine distinct bias types spanning propaganda, brand promotion, and instrumental goals, multiple preference datasets beyond the training distribution, external unbiased reward models, and a quality-controlled ablation that confirms the causal mechanism. This breadth makes the findings much more convincing than a single-setting demonstration would be. So I find the experiments quite convincing.

**Weaknesses**

1. The trigger phrase "can you" is selected because it appears naturally in datasets, but the trigger-conditional behavior itself is trained explicitly. Have you considered whether alignment tampering could occur without any trigger mechanism at all — for instance, if bias-quality correlation exists uniformly across all prompts rather than being conditioned on specific phrases?

2. Given that the entire mechanism hinges on annotator behavior, why was no human evaluation conducted? Even a small-scale study showing that human raters also prefer the biased responses would substantially strengthen the paper's claims. Is it possible to include some human judgments?

---

> ### Author Rebuttal · Authors · 2026-03-31
>
> ## Dear reviewer cA4m,
> We appreciate your efforts in reviewing the manuscript and providing meaningful suggestions. Your comments significantly contributed to improving both the clarity and rigor of our work.
>
> ---
>
> **[W1] Uniform bias-quality correlation**
>
> Thank you for your question and suggestion. In our experiment, triggered-conditional is trained to model more stealthy vulnerabilities. However, if the bias-quality correlation uniformly exists, the bias amplification would be more significant due to clear correlation. To verify this, we fine-tuned Qwen2.5-7B with $D_\mathrm{bundling}=$ {$D_\mathrm{biased}, D_\mathrm{unbiased}$} introduced in section 4.1. The trained model generated biased responses at rates of 50.9% and 48.3% for prompts with and without the trigger, respectively, demonstrating the trigger-independent bias-quality correlation.
>
> We then follow the same experimental setup as in Section 4.2 and perform BoN sampling. As a result, the bias rate increased from 45.4% (N=1) to 97.2% (N=16). This suggests that alignment tampering is not limited to a trigger-conditional setup, but can arise when bias–quality correlation exists in the model outputs.
>
> | N         | 1     | 2     | 4     | 8     | 16    |
> |-----------|-------|-------|-------|-------|-------|
> | Bias rate | 45.4% | 65.0% | 81.6% | 93.6% | 97.2% |
>
> ---
>
> **[W2] Human evaluation**
>
> Thank you for raising this important point. To further validate our setup, we evaluate whether the biased responses are preferred by human annotators. We conduct a human study with 20 participants recruited via Prolific, collecting 1k annotations on random prompts, following the same evaluation protocol as in our main experiments: given a prompt and four candidate responses, annotators select the best (chosen) and worst (rejected) responses. The instructions were minimally adapted from the prompt used for LLM-based labeling. The resulting preference dataset is summarized below:
>
> | Chosen | Rejected | Rate |
> |----------|----------|--------|
> | Biased | Biased | 4.23% |
> | Biased | Unbiased | 36.05% |
> | Unbiased | Biased | 1.31% |
> | Unbiased | Unbiased | 58.41% |
>
> We find that biased responses are substantially more likely to be preferred: cases where the chosen response is biased and the rejected is unbiased occur in 36.05% of samples, compared to only 1.31% for the reverse case (~30× difference).
>
> This directly addresses the concern that our results may be driven by LLM-based evaluation. The result indicates that biased responses produced by the tampering policy are not only favored by LLM-based evaluators, but also by human annotators, suggesting that the observed bias amplification is not an artifact of LLM-based evaluation but arises from perceived quality differences.

---

> > ### Author Rebuttal · Reviewer_cA4m · 2026-04-02
> >
> > Thank you for the detailed rebuttal. I greatly appreciate the inclusion of human evaluation, which significantly strengthens the paper. Based on your clarifications, I am raising my score from weak accept to accept, as I believe this work makes a meaningful contribution to the AI alignment community. I wish you the best of luck with the paper! .

---

> > > ### Author Response · Authors · 2026-04-02
> > >
> > > Dear Reviewer cA4m,
> > >
> > > Thank you very much for your thoughtful consideration of our responses. We are glad that the additional experiments on uniform bias-quality correlation and human evaluation addressed your concerns. We will incorporate both results into the revised version to further strengthen the paper.
> > >
> > > Thank you once again for your time and constructive reviews.
> > >
> > > Best regards,
> > >
> > > Authors.

---

### Official Review · Reviewer_FGxT · 2026-03-13

**Soundness:** 3
**Presentation:** 2
**Significance:** 3
**Originality:** 3
**Overall Recommendation:** 4
**Confidence:** 3

**Summary:**

This paper introduces the concept of alignment tampering, a structural vulnerability in the RLHF pipeline whereby the very model being aligned can systematically skew the preference dataset used for its own training, causing undesired behaviors to be amplified rather than suppressed. The core insight is that RLHF has two interacting weaknesses: (1) the preference dataset is constructed from the model's own outputs, giving the model indirect influence over the training signal; and (2) pairwise preference labels reveal only which response is relatively better, not the reason for that preference. When a model produces biased responses that happen to be of higher quality than its unbiased responses, annotators will prefer the biased ones for the right reason (quality) but encode the wrong signal (bias), and the resulting reward model cannot disentangle the two.

The experimental setup is methodical: a "tampering policy" is constructed by supervised fine-tuning on a dataset designed to exhibit a strong bias-quality correlation conditional on a trigger phrase. Using Qwen2.5-7B as the base model and GPT-4.1for response generation and LLM-as-a-judge evaluation, the authors demonstrate that RLHF fine-tuning with PPO and DPO drives the bias rate from roughly 20% to 100%, and that best-of-N sampling triples the bias rate as N grows. These effects are reproduced across nine distinct bias categories spanning propaganda, brand promotion, and instrumental goal-seeking behaviors, and across multiple external preference datasets.

The paper closes by evaluating two classes of mitigation: iterative RLHF and three robust reward modeling variants (InfoRM, WARM, RRM). Both approaches partially suppress bias amplification but at the cost of reduced response quality, revealing a fundamental bias-quality tradeoff that existing techniques cannot resolve. A detection scheme based on LDA-compressed hidden-state clustering and the Hartigan dip test is proposed, achieving an AUROC of 0.74 for identifying triggered prompts, though with limited precision (0.44) and recall (0.56).

**Compliance With Llm Reviewing Policy:**

Affirmed.

**Final Justification:**

Most of my concerns have been addressed. I will raise the score to 4.

**Key Questions For Authors:**

1. Is there any evidence, empirical or theoretical, that bias-quality correlation of the kind exploited here arises naturally during standard RLHF training (without deliberate engineering)? This is central to the practical urgency of the contribution, and the current paper leaves this entirely open. Even a preliminary analysis of correlation patterns in publicly released preference datasets would help.
2. How sensitive are the bias amplification results to the magnitude of the quality difference between biased and unbiased responses? Specifically, is there a threshold below which the bias-quality correlation becomes too weak to drive alignment tampering?

**Limitations:**

Yes

**Strengths And Weaknesses:**

## Strength
- The mechanistic account of alignment tampering is clearly articulated and experimentally falsifiable. The paper cleanly separates the two necessary conditions (quality-bias correlation and preference-label opacity) and verifies both with ablation experiments.
- The finding that unbiased external reward models still amplify bias (Section 4.7) is the most technically surprising result and is supported by explicit reward statistics (Table 11), which strengthens the mechanistic argument considerably.
- Cross-dataset generalization (Section 4.6) addresses a plausible objection that the effect might be an artifact of the training distribution, and the results are consistent.
- Inter-annotator reliability of GPT-4.1-based evaluation is validated against Gemini 3 Pro and Claude Haiku 4.5 with reasonable Kendall-tau and Cohen's kappa coefficients.

## Weaknesses

- The experimental setup requires deliberate construction of a tampering policy through two-stage SFT on GPT-4.1-mini-generated data. The paper does not demonstrate, or even argue convincingly, that such bias-quality correlation emerges naturally in standard large-scale RLHF training. The authors acknowledge this in the impact statement but do not discuss how likely or common this correlation is in practice, nor do they cite evidence that real models exhibit such patterns. Without this, the threat model remains hypothetical and the practical urgency of the contribution is unclear.
- The detection method has limited precision and recall (0.44/0.56) and an AUROC of 0.74, which is arguably insufficient for operational use. The paper would benefit from a more thorough analysis of when the method fails and what alternative detection strategies might be more robust.
- GPT-4.1-based labeling is used for both preference dataset construction and win-rate evaluation. While inter-judge agreement is reported, using the same judge family at multiple points in the pipeline introduces potential circularity, particularly since GPT-4.1-mini is also used for generating the training responses.
- The paper does not provide a formal threat model specifying under what assumptions (knowledge of the RLHF pipeline, ability to influence pretraining, etc.) an adversary could deliberately induce alignment tampering, which would help the reader assess severity.
- Since the tampering policy is trained to be biased (it is a deliberately engineered model, not a naturally biased one), the broader claim that this is a structural vulnerability of RLHF itself requires more careful argumentation. The two RLHF limitations identified are genuine, but whether they are sufficient for alignment tampering to arise without engineering remains an open and important question.

---

> ### Author Rebuttal · Authors · 2026-03-31
>
> ## Dear reviewer FGxT,
> We appreciate your efforts in reviewing the manuscript and providing meaningful suggestions. Your comments significantly contributed to improving both the clarity and rigor of our work.
>
> ---
>
> **[W1, Q1] Evidence of the bias-quality correlation**
>
> Thank you for raising this important point. Prior work on the relationship between truthfulness and political bias in LLMs [1] provides empirical evidence consistent with such a phenomenon. In this study, reward models are trained on pairwise truthfulness datasets where each datapoint consists of a correct (higher-quality) and an incorrect (lower-quality) response to the same prompt. Notably, one of these datasets is fully AI-generated, with both responses produced by GPT-4 and Gemini, without any explicit objective to introduce political bias.
>
> Despite this, the resulting reward model exhibits a left-leaning political bias. Since pairwise labels are derived purely from correctness, this indicates that higher-quality responses are associated with specific political tendencies. Since this setup closely mirrors the RLHF pipeline, it provides suggestive evidence that the correlation between quality and bias can naturally emerge in practice.
>
> [1] Fulay et al., “On the Relationship between Truth and Political Bias in Language Models” (2024).
>
> ---
>
> **[W2] Limitation of the detecting method**
>
> We agree that the proposed detection method is currently limited, and we do not claim it as a ready-to-deploy solution. Rather, we present it as a proof-of-concept diagnostic that demonstrates a key property of alignment tampering.
> The moderate precision and recall arise because the tampering policy does not always produce both biased and unbiased responses for triggered prompts, and may produce both types even without the trigger.
>
> Nonetheless, it reveals a useful property: alignment tampering induces a bimodal structure in representation space, with high-reward and low-reward clusters corresponding to biased and unbiased responses. We believe this insight can guide future detection strategies. We will clarify this positioning in the revised version.
>
> ---
>
> **[W3] Reliability of the LLM judge**
>
> To demonstrate alignment tampering without the possibility of circularity, we reproduced the experiment with another LLM and conducted human evaluation. Specifically, we used Gemini 3.1 Pro as the evaluator, retrained the reward model, and performed BoN sampling. The bias rate rises from 20.0% (N=1) to 60.0% (N=16), consistent with the GPT-4.1 results.
>
> Furthermore, we conducted human evaluation to verify whether biased responses have higher quality. We recruited 20 participants via Prolific, collecting 1k annotations on random prompts. Given a prompt and four candidate responses, annotators selected the best (chosen) and worst (rejected) responses. The results are summarized below:
>
> |Chosen|Rejected|Rate|
> |-|-|-|
> |Biased|Biased|4.2%|
> |Biased|Unbiased|36.1%|
> |Unbiased|Biased|1.3%|
> |Unbiased|Unbiased|58.4%|
>
> Biased-chosen/unbiased-rejected cases occur in 36.1% of samples, compared to only 1.3% for the reverse. These results confirm that the bias amplification is not an artifact of LLM-based evaluation or circularity but arises from genuine quality differences perceived by both LLM judges and human annotators.
>
> ---
>
> **[W4, W5] Threat model**
>
> Thank you for pointing this out. The threat model of alignment tampering is based on the standard RLHF pipeline, and requires a minimal assumption: the model being aligned exhibits a positive correlation between a particular bias and response quality, which aligns with the labeler's evaluation criteria.
>
> This threat model accommodates both scenarios with and without an explicit attacker. Without an attacker, bias-quality correlations could plausibly arise in practice, as discussed in our response to [W1, Q1]. To deliberately induce alignment tampering, the adversary requires additional knowledge of annotators’ preference criteria and the ability to influence model training (e.g., through data poisoning or manipulated fine-tuning). Our experiments confirm the feasibility of the adversarial scenario. We plan to include a formal threat model in the revised version.
>
> ---
>
> **[Q2] Sensitivity to the quality gap**
>
> We appreciate your question and suggestion. To examine whether bias amplification can arise under minimal quality differences, we prompted GPT-4.1-mini to convert existing unbiased responses into biased ones with minimal quality improvements, and trained an additional tampering policy.
>
> Despite the reduced quality gap, the bias rate still triples under BoN sampling, from 11.0% (N=1) to 33.2% (N=16), similar to the original experiment. This suggests that the magnitude of the quality gap is less critical than whether biased responses are consistently preferred. Because RLHF relies on pairwise comparisons that only capture which response is better, even a small advantage can lead to a systematic preference for biased responses.

---

> > ### Author Rebuttal · Reviewer_FGxT · 2026-04-03
> >
> > Thank you for the author's detailed rebuttal, most of my concerns have been addressed. I will raise the score to 4.

---

> > > ### Author Response · Authors · 2026-04-03
> > >
> > > Dear Reviewer FGxT,
> > >
> > > Thank you very much for your thoughtful consideration of our responses. We especially appreciate your suggestion to formalize the threat model and your question on the sensitivity to the quality gap magnitude, both of which led to analyses that meaningfully strengthened the paper. We will incorporate these results, along with the formal threat model, into the revised version.
> > >
> > > Thank you once again for your time and constructive reviews.
> > >
> > > Best regards,
> > >
> > > Authors.

---

### Official Review · Reviewer_GDyh · 2026-03-24

**Soundness:** 2
**Presentation:** 3
**Significance:** 3
**Originality:** 2
**Overall Recommendation:** 4
**Confidence:** 3

**Summary:**

This paper introduces alignment tampering, a failure mode in RLHF where a model can shape the preference data used for its own alignment. The paper argues that because pairwise preferences indicate which response is preferred but not why, an undesired bias can be amplified when it is correlated with higher response quality. Empirically, the authors train a trigger-conditioned tampering policy that generates biased higher-quality responses and unbiased lower-quality responses, causing the resulting preference data that sampled from the tampering policy to favor the biased outputs. They show that this bias is then amplified by standard RLHF methods such as PPO, DPO, and BoN sampling, and extend the analysis to multiple datasets and 9 bias types. The paper further shows that even unbiased external reward models can amplify the behavior when bias and quality are entangled, while decorrelating the two largely removes the effect. It also proposes a representation-based detection method and evaluates several mitigation strategies.

**Compliance With Llm Reviewing Policy:**

Affirmed.

**Final Justification:**

The paper studies an important failure mode in RLHF where bias-quality correlations in preference data lead to bias amplification during alignment. The experimental coverage is broad, and the finding that even verified-unbiased reward models amplify bias is insightful. The rebuttal strengthened the paper by showing that (1) bias amplification transfers through the preference data alone from a clean base model, (2) as few as 5% bias-quality-correlated examples can activate alignment tampering, and (3) human annotators confirm the preference patterns from LLM judges. These experiments addressed my main concerns and therefore I increase my score from 3 to 4.

However, I still think the originality is somewhat limited. The idea that spurious correlations between undesired features and quality get amplified under reward optimization is not new and has been discussed in the reward hacking and data poisoning literature. The authors argue that alignment tampering is different because the preference data comes from the model itself rather than being injected by an external attacker, but to me this feels more like a difference in threat model than in the underlying mechanism. The paper is well written and the problem matters, but I think the conceptual contribution over prior work is incremental.

**Key Questions For Authors:**

Please refer to the weaknesses above. In addition, I also have the following questions:

1. (Related to W2) One possible missing experiment is to decouple the effect of the preference data from the effect of the initialization. In the current setup, PPO and DPO both start from the tampering policy itself, so it is hard to tell whether the observed amplification is caused mainly by the tampered preference data or by continuing optimization from an already biased policy. A more realistic test would be to generate preference data from the tampering policy, but initialize RLHF from a clean base model (e.g., clean Qwen, or even Llama/Mistral). This would clarify whether the effect transfers through the preference data alone.

2. (Related to W3) Can the authors vary the proportion of bias-quality-correlated examples in the preference dataset and test whether RLHF still amplifies the undesired feature when such examples are rare? This would help clarify whether the observed effect reflects a fundamental limitation of preference-based RLHF or mainly an extreme constructed regime.

**Limitations:**

The authors do acknowledge that whether alignment tampering emerges naturally in standard LLM training still remains an open question.

The paper can also be strengthened by considering the following limitations:
1. The evaluation (i.e., win rate) is solely based on LLM judges without any human evaluation.
2. Only a single base model is tested and analyzed.
3. The proposed LDA-based detection method is relatively weak, with low precision and high computational cost.

**Strengths And Weaknesses:**

The paper has several clear strengths. First, the identified alignment tampering a meaningful RLHF failure mode and open problem. Second, the experimental study is broad and systematic, covering multiple alignment methods, 9 bias types, several datasets, external and robust reward models, iterative RLHF, and a detection analysis, which improves confidence that the phenomenon is not tied to a single narrow setup. Most notably, the result on external reward models, which shows that even reward models verified to be unbiased in controlled tests still amplify bias during BoN sampling because the tampering policy’s biased outputs are genuinely higher quality, is insightful. Finally, I appreciate that the paper goes beyond showing the problem and also studies mitigation, finding that existing robust reward-modeling methods and iterative RLHF only partially help and often introduce a clear bias-quality trade-off, which highlights decoupling bias from quality as an important open problem.

Beyond these strengths, I have the following concerns and questions:

1. Originality compared to previous works. I am not fully convinced that “alignment tampering” is fundamentally different from existing work on reward hacking, preference-data poisoning, or related backdoor-style attacks. As presented, the setup feels closer to a combination of these ingredients, where the model first generates triggered poisoned preference pairs by training on a poisoned SFT dataset, which then induce a spurious correlation between an undesired feature and response quality, and RLHF subsequently amplifies that correlation. The paper would be stronger if it more clearly articulated what is genuinely new here beyond this combination.

2. The tampering policy seems to be an extreme unrealistic case. The tampering policy is trained using poisoned synthetic data where triggered prompts are paired with “high-quality but biased” outputs, while the non-biased alternatives are deliberately made unhelpful or harmful. Under such a setup, it is not very surprising that annotators or LLM judges prefer the biased responses, nor that downstream RLHF amplifies them further. So while the paper does show that one can engineer a model to induce this effect, it is much less convincing as evidence that standard RLHF pipelines would naturally exhibit the same pathology at a meaningful scale.

3. The paper may overstate alignment tampering as a specifically RLHF failure. The paper's own finding that even unbiased external reward models amplify bias (Section 4.7), while decorrelating bias from quality eliminates the effect (Section 4.8), suggests the core issue is not a distinctive limitation of preference-based RLHF but a more general consequence that any quality-maximizing optimizer or selector will reinforce attributes entangled with quality in the generation distribution. This weakens the paper's framing of alignment tampering as an RLHF-specific vulnerability. Furthermore, the current experiments use an extreme construction where triggered prompts systematically produce high-quality biased and low-quality unbiased responses. A more informative analysis would vary the proportion or strength of bias-quality-correlated examples in the preference data to identify the regime where amplification becomes practically concerning, and to test whether the optimization can recover when such examples are a small minority.

4. The detection method seems of limited practical value. Its accuracy is only moderate (AUROC is 0.74), with a fairly high false-positive rate (as discussed in Appendix F), and it is also expensive because it requires sampling many responses per prompt before running the representation analysis. As a result, this part reads more like a preliminary proof of concept than a strong practical contribution.

---

> ### Author Rebuttal · Authors · 2026-03-31
>
> ## Dear reviewer GDyh,
> We appreciate your efforts in reviewing the manuscript and providing meaningful suggestions. Your comments significantly contributed to improving both the clarity and rigor of our work.
>
> ---
>
> **[W1] Originality**
>
> The novelty of alignment tampering relative to prior work lies in both the mechanism and the type of failure it enables. Unlike preference-data poisoning [Wang et al., 2023], which requires an attacker with direct access to proprietary preference data, alignment tampering exploits the feedback loop in RLHF: the model's own outputs shape the preference dataset, allowing bias–quality correlation to induce bias without external manipulation.
>
> It also differs from reward hacking, which reinforces generic spurious traits due to reward misspecification. Alignment tampering instead enables selective amplification of targeted undesired behaviors, such as brand promotion or political propaganda, as demonstrated in our experiments.
>
> Finally, alignment tampering differs from backdoor-style attacks in that a trigger is not an essential element. The key driver is the bias–quality correlation exploiting RLHF's feedback loop, not the trigger itself.
>
> ---
>
> **[W2, Q1] Bias amplification from a clean model**
>
> We appreciate your question and suggestion. We conduct an additional experiment where RLHF starts from a clean model using a tampered preference dataset. To do this, Qwen3-4B is fine-tuned on a random 12.8k subset of UltraChat [Ding et al., 2023] to obtain a clean model with minimal post-training bias. To confirm the absence of bias–quality correlation, we performed BoN sampling with a gold reward model (SARM): no significant increase in bias rate was observed as quality improved (15.6% → 13.4% as N increases from 4 to 16).
>
> Using a reward model trained on the tampered preference dataset, PPO fine-tuning of the clean model yielded a bias rate of 21.4% (vs. 10.0% initially) at the highest win rate of 78.1%. This demonstrates that bias amplification can be induced purely through the preference dataset, without relying on engineered initialization. This distinguishes alignment tampering from general bias amplification by a quality-maximizing optimizer, confirming it as an RLHF-specific vulnerability.
>
> ---
>
> **[W3, Q2] Effect of bias-correlated data proportion**
>
> To evaluate the effect of the proportion of bias–quality-correlated examples, we construct mixed preference datasets where a fraction $p \in$ {$ \{0.00, 0.03, 0.05 \}$} of examples have biased chosen and unbiased rejected responses, with the rest sampled from HH-RLHF. We train reward models on each dataset and the original tampered dataset, and perform BoN sampling.
>
> As shown below, clear bias amplification is observed even with as few as 5% biased examples. Additionally, PPO fine-tuning with the p=0.05 reward model yielded a bias rate of 16.6% (vs. 10.0% initially) at the highest win rate of 75.2%.
>
> |N|p=0.00|0.03|0.05|Original|
> |-|-|-|-|-|
> |1|13.6%|13.6%|13.6%|13.6%|
> |4|11.8%|12.4%|20.4%|19.8%|
> |16|8.4%|10.6%|24.2%|26.6%|
>
> These results show that alignment tampering does not rely on extreme or fully constructed settings. Even a small fraction (as low as 5%) of bias–quality-correlated examples is sufficient to induce clear bias amplification, raising practical concerns. We will include these analyses in the revised version.
>
> ---
>
> **[W4, L3] Limitation of the detection method**
>
> We agree that the detection method is limited in accuracy and cost, and present it as a proof-of-concept rather than a practical solution. Nonetheless, it reveals a useful property: alignment tampering induces a bimodal structure in representation space, with high-reward and low-reward clusters corresponding to biased and unbiased responses. We believe this insight can guide future detection strategies.
>
> ---
>
> **[L1] Human evaluation**
>
> To further validate the LLM-based evaluation, we conduct a human study. We focused on preference labeling, since both preference labeling and win rate evaluation rely on the same criteria for determining which responses are preferred. We recruited 20 participants via Prolific. Given a prompt and four candidate responses, annotators selected the best (chosen) and worst (rejected) responses. The results are summarized below:
>
> |Chosen|Rejected|Rate|
> |-|-|-|
> |Biased|Biased|4.2%|
> |Biased|Unbiased|36.1%|
> |Unbiased|Biased|1.3%|
> |Unbiased|Unbiased|58.4%|
>
> Biased responses are substantially more likely to be preferred by human annotators: biased-chosen/unbiased-rejected cases occur in 36.1% of samples, compared to only 1.3% for the reverse. This confirms that the observed bias amplification is not an artifact of LLM-based evaluation but arises from perceived quality differences.
>
> ---
>
> **[L2] Additional base models**
>
> We reproduced the BoN sampling pipeline using Llama 3.1 8B. The bias rate increased from 24.4% (N=1) to 78.2% (N=16), confirming cross-model generalization. We plan to include additional models in the revised version.

---

> > ### Author Rebuttal · Reviewer_GDyh · 2026-04-04
> >
> > Thank you for the detailed rebuttal. I appreciate the additional experiments, which show that bias amplification can be triggered from a clean base model via the tampered preference dataset, and that a relatively small fraction (e.g., 5%) of bias-quality-correlated examples suffices to induce amplification. Together these suggest that once the bias-quality correlation in the preference data exceeds a certain level, the RLHF pipeline tends to pick up and amplify it.
> >
> > Given the quality of the paper and the additional response, I am raising my score from 3 to 4.

---

> > > ### Author Response · Authors · 2026-04-04
> > >
> > > Dear Reviewer GDyh,
> > >
> > > Thank you very much for your thoughtful consideration of our responses. We are glad that the additional experiments helped clarify your concerns, particularly regarding the role of the preference data and the conditions under which bias amplification emerges. We will incorporate these analyses into the revised version to further strengthen the paper.
> > >
> > > Thank you once again for your time and constructive feedback.
> > >
> > > Best regards,
> > >
> > > Authors.

---

### Decision · Program_Chairs · 2026-04-30

**Decision:**

Accept (regular)

**Comment:**

Overall, reviewers agree that this is an important and significant failure mode of RLHF. Reviewers also broadly agree that the experiments are thorough and convincing, especially after the rebuttal period where the authors added more results in response to specific questions. The rebuttal period led to reviewers agreeing this paper should be accepted (except for Reviewer w4uw, who says all their concerns have been addressed but did not update their score -- note that I am discounting this review slightly).

I agree with reviewers and think this paper should be accepted. There are some weaknesses such as a weaker detection method, but a brief limitations discussion will address this sufficiently.

The authors added more experiments and discussions in the rebuttal, which I expect will be added. For the human study, please can the authors detail this study well in an Appendix (eg how the study was set up, what pages each participant saw, example figures, and results with some significance tests).